# Quality of Life of Allergic Dogs Treated with Allergen-Specific Immunotherapy—A Retrospective Study

**DOI:** 10.3390/vetsci10020072

**Published:** 2023-01-18

**Authors:** Tina Kotnik

**Affiliations:** Small Animal Clinic, Veterinary Faculty, University of Ljubljana, 1000 Ljubljana, Slovenia; tina.kotnik@vf.uni-lj.si; Tel.: + 38-6-1-4779-284

**Keywords:** quality, ASIT, immunotherapy, dogs, canine, atopic, dermatitis, CAD

## Abstract

**Simple Summary:**

The goal of veterinary intervention in allergic dogs is to improve their quality of life, which should be as comparable as possible to that of healthy dogs. Allergen-specific immunotherapy is an effective treatment in which allergic dogs are administered an allergen mixture to reduce or eliminate symptoms associated with subsequent exposure to the causative allergens. The study examined the quality of life of dogs receiving allergen-specific immunotherapy compared with dogs treated with other therapies. The quality of life of dogs treated with allergen-specific immunotherapy improved significantly. The dogs were significantly less disturbed at mealtimes (i.e., had better appetites and scratched less at mealtimes) and caused significantly less physical discomfort to their owners (from unpleasant odors to the impression of a dirty apartment) than dogs not treated with allergen-specific immunotherapy. Owners of dogs that received allergen-specific immunotherapy were able to significantly improve their daily activities (such as leisure, vacation, walks, work, hunting), significantly reduce their expenses (treatment costs, veterinary costs), feel significantly less emotional distress (less guilt, powerlessness, sadness, regret, fear, anger, disgust, rage, frustration), and experience less influence on relationships with family members and friends compared to owners of dogs that were not treated with allergen-specific immunotherapy. By using quality-of-life questionnaires, veterinarians can demonstrate improvements during therapy to gain owner confidence.

**Abstract:**

Background: The quality of life (QoL) of dogs with canine atopic dermatitis (CAD) treated with allergen-specific immunotherapy (ASIT) was studied to determine whether the QoL of dogs treated with ASIT (AG) improved compared with dogs not treated with ASIT (CG). Methods: The power of the study was calculated in advance assuming that the AG would assess QoL 20% better than the CG. The CG consisted of 21 dogs with CAD and the AG of 46 dogs with CAD. Validated QoL1 (pre-treatment) and QoL2 (post-treatment) questionnaires were emailed to owners. Results: AG dogs were significantly less disturbed during mealtimes (i.e., had better appetites and scratched less during mealtimes) and caused significantly less physical discomfort to their owners (due to unpleasant odor, the impression of a dirty apartment) than CG dogs. Owners of dogs treated with ASIT were able to significantly improve their daily activities (leisure, vacation, walks, work, hunting), significantly reduce their expenses (treatment costs, veterinary costs), feel significantly less emotional distress (less guilt, powerlessness, sadness, regret, fear, anger, disgust, rage, frustration), and experience less influence on relationships with family members and friends compared to owners of dogs not treated with ASIT. Conclusions: According to our results, the quality of life of dogs treated with ASIT and their owners seemed to improve significantly.

## 1. Introduction

A group of veterinary dermatologists have developed and validated a quality-of-life (QoL) questionnaire for the owners of dogs with skin disease [1,2]. This type of questionnaire can be used by clinicians on a daily basis as a tool to evaluate different therapeutic interventions in allergic animals. Clinicians who use these tools can not only receive relevant feedback on their interventions but also improve dog owner compliance by involving them in their animal treatment. By being able to show an improvement in quality of life during treatment (by comparing owners’ responses at different times during therapy), the clinician can more easily gain owners’ devotion [3,4].

“Quality of life” (QoL) is generally defined as “the degree to which an individual enjoys his or her life”. In veterinary medicine, QoL has been defined as “the level of an individual’s satisfaction (needs and desires) that are determined by the individual’s living conditions, which then determine factors such as health, happiness and longevity” [5].

Canine atopic dermatitis (CAD) has been defined as a genetically predisposed inflammatory and pruritic allergic skin disease with characteristic clinical features [6]. Interventions that address pruritus in allergic dogs seem to be the most important in improving the QoL of the animals as well as the owners [7]. Among antipruritic drugs, symptomatic medicaments are one option. Effective drugs such as glucocorticoids and cyclosporin-A often cause undesirable side effects. Novel therapeutics such as monoclonal antibodies against specific cytokines are effective but quite expensive [7,8,9]. Regardless of which drug we use, symptoms are only relieved during the period of treatment.

A treatment that may alleviate symptoms in allergic animals for longer periods compared to symptomatic drugs is allergen-specific immunotherapy (ASIT). Allergen-specific immunotherapy is defined as the practice of administering gradually increasing quantities of allergen extracts to an allergic patient; the purpose of this is to reduce or eliminate the symptoms associated with subsequent exposures to the causative allergens [10]. It is the only current allergy treatment that can alter or reverse at least part of the immune response in this condition, thereby both relieving clinical symptoms and preventing disease progression. This modification is accomplished without the possible long-term adverse effects of a lifetime of drug treatment, with minimal adverse effects, and with the potential of long-lasting effectiveness [11,12].

The quality of life of both allergic dogs treated with ASIT and of their owners has not yet been evaluated. The primary outcome of this study was to determine if the QoL of dogs and owners improved as a result of ASIT. The secondary outcome was to determine if the QoL of ASIT-treated dogs improved compared to dogs not treated with ASIT.

## 2. Materials and Methods

Dogs with CAD treated with various symptomatic drugs and ASIT were included in this study (ASIT Group; AG). The control group (CG) consisted of dogs with CAD treated with various symptomatic drugs but not ASIT. The drugs that were allowed to be used in both groups were topical and systemic antipruritics, antimicrobials, and antiparasitic treatments. Dogs that may have had systemic or severe illnesses other than CAD, which could have impaired QoL, were not included.

Medical records of the Small Animal Clinic of the Faculty of Veterinary Medicine in Ljubljana were searched, and dogs with CAD were selected. The diagnosis was based on guidelines and good clinical practice [13]. In short, a clinical diagnosis has been made after the exclusion of ectoparasitoses based on negative skin scrabs and diagnostic therapy with broad-spectered ectoparasiticides. Secondary bacterial and/or fungal infections have been excluded or treated prior to the definite diagnosis, based on cytology and/or microbiology results. Food allergies (in the cases of concomitant disease) have been controlled with an appropriate, individually tailored, homemade elimination diet.

Serum samples from dogs included in AG were tested for IgEs against environmental allergens by ELISA at the reference laboratory (Alergovet Laboratories, Valentín Beato 244ta planta, oficina 8B28037, Madrid, Spain). According to the guidelines, the test results were used to identify the triggering allergen(s) to develop allergen-specific immunotherapy (ASIT) [13,14]. The blood of dogs enrolled in the CG was not tested.

Validated questionnaires assessing the quality of life (QoL) of dogs with skin disease and their owners [1,2] were translated into Slovenian and sent by email to the owners of dogs selected for enrollment. Each owner completed 2 questionnaires. The first was related to the dog’s quality of life before treatment and the second to the dog’s quality of life for at least 1 year after the start of treatment. Owners were also asked about the duration of ASIT treatment and whether it had already been discontinued, and, if so, what the reason was for discontinuation. Owners gave written consent to record the data and were assured that no identification would be possible due to complete anonymization.

The allergen extract used for ASIT was manufactured by Nextmune (Vijzelweg 11, 8243 PM Lelystad, The Netherlands) and administered subcutaneously. All participating dogs were prescribed the same desensitization protocol (Nextmune official instructions). Briefly, the protocol consisted of s/c injections of allergen mixtures increasing from 0.2 mL to 1.0 mL. Therapy was started with 4 injections every 2 weeks, continued with 2 injections every 3 weeks, and maintained with 1 injection every 4 weeks. Individual adjustments to the dose were possible if the physician determined intolerance. The duration of therapy under study was not limited as long as reactions were cleared. The composition was individual for each patient. A determination of whether the dogs were actually exposed to the allergen(s) to which they reacted was made in a personal interview with the owners before the drug was compounded and in conjunction with the clinical history and clinical picture. When necessary, symptomatic medications were allowed to be administered concurrently in the AG. The dogs of the CG were treated with various symptomatic medications but not ASIT.

The sample size was estimated in advance. Using a free online calculator, we determined the sample size (http://powerandsamplesize.com, page last accessed 12 September 2022). We assumed that owners of the dogs treated with ASIT would report an estimated 20% better QoL than owners of the dogs not treated with ASIT, with a standard deviation of 20%. Using these values, we determined that we needed to include at least 18 dogs per group for this study to have 95% power to detect a significant difference in QoL between groups at *p* ≤ 0.05.

The computer program Python and its module scipy.stats (https://docs.scipy.org/doc/scipy/reference/stats.html, accessed on 29 August 2022) were used for data analysis. The Shapiro–Wilk test (scipy.stats.shapiro) was used to determine data distribution for each question separately. Based on the findings, parametric tests or nonparametric tests were used to compare data among groups of dogs. Accordingly, a paired two-tailed *t*-test (scipy.stats.ttest_rel) with a significance level of 0.05 was performed for each question on pretreatment and posttreatment responses for each group separately (for normally distributed data) or the Wilcoxon signed-rank test (scipy.stats.wilcoxon) was used (for non-normally distributed data of at least one set of responses for each question). To compare the ASIT-treated dogs and control dogs, an unpaired two-tailed *t*-test (scipy.stats.ttest_ind for normally distributed data) or the Mann–Whitney test (scipy.stats. mannwhitneyu for non-normally distributed data of at least one set of responses for each question) with a significance level of 0.05 was performed for each question.

## 3. Results

### 3.1. Included Animals

Between September 2021 and February 2022, 66 emails were sent to owners of dogs to be enrolled in the CG and 66 emails were sent to owners of dogs to be enrolled in the AG, following invitations issued by phone. The control group consisted of 21 responders and the AG of 46 responders. The median age of the dogs in the CG was 7 years (min 2; max 14 years) and the median age of the dogs in the AG was 8 years (min 2; max 15 years) at the time of inclusion.

### 3.2. Epidemiology of ASIT Treatment

At the time of inclusion, immunotherapy had already been finished in a period ≤10 months in 12 cases (26.1%). Treatment was discontinued by veterinary or owner decision because the clinical condition had improved in 5 of 12 dogs (41.7%) and worsened in the others (15.2%). The dogs in which ASIT was discontinued regardless of response were included in the AG. Treatment has been continued for >10 months in 34 cases (73.9%). Due to unsatisfactory results, an additional 12 dogs (26.1%) discontinued treatment after a median treatment duration of 1.5 years. Among long-treated dogs, 22 dogs (47.8%) were still on the treatment at the time of enrollment with a median time of the treatment of 3 years. Thus, a long-lasting and satisfactory response to ASIT was achieved in 58.7% of treated dogs.

### 3.3. Quality-of-Life Questionnaires Results

Owners completed two quality-of-life questionnaires: QoL1 (before treatment) and QoL2 (after treatment). Owners were instructed to answer QoL1 first and then QoL2. For each question on QoL1, “before treatment” was written, and for each question on QoL2, “after treatment” was written. Only one owner answered the questionnaires for each dog, and only the dogs of owners who completed all questions and returned both questionnaires were included in CG and AG. The results of the CG are shown in Table 1 and the results of the AG group are shown in Table 2.

A statistical comparison of responses to QoL1 and QoL2 within CG showed significant differences in questions 1, 2, 3, 4, 5, 7, 11, 13, and 15 (Table 1 and Figure 1). The various symptomatic treatments significantly improved the quality of life of the dogs and their owners in many areas of clinical presentation and behaviour, e.g., symptoms were significantly less severe, and the dogs were less disturbed, in a better mood, less lethargic, less nervous and aggressive, or less restrained than before treatment. The dogs’ sleep, meals, work, and play activities were significantly less disturbed by scratching and licking than before treatment. Owners felt less guilt, powerlessness, sadness, regret, anxiety, anger, disgust, rage, and frustration, and the disease had significantly less impact on relationships between family members than before treatment.

However, despite treatment, the disease still caused abnormal interactions between the dogs and their owners, physical discomfort, inconvenience and significant loss of time due to frequent shampooing, the administration of tablets, and ear cleaning (questions 6, 8, 9, and 14, respectively). Owners felt as tired as they did before treatment, and the impact on their spending remained the same (questions 10 and 12, respectively). Owners’ ordinary activities, such as leisure, holidays, walking, working, and hunting, were significantly more affected by the dogs’ illness after treatment than before treatment (question 11).

A statistical comparison of responses to QoL1 and QoL2 within AG (Table 2) showed significant differences for all questions except question 8 (Figure 2). Under ASIT, the dogs were shampooed, treated with tablets, and had their ears cleaned to a comparable extent as before treatment. Otherwise, the quality of life of the dogs and their owners was significantly better under ASIT than before treatment for all determinants.

Finally, a statistical comparison of responses to QoL2 was performed between the CG and the AG (Figure 3). It was found that dogs treated with ASIT were significantly less disturbed during meals (i.e., had better appetites and scratched less during meals) and caused significantly less physical discomfort in their owners (due to unpleasant odor, the impression of a dirty apartment) than dogs not treated with ASIT (questions 4 and 14, respectively). Owners of dogs treated with ASIT were able to significantly improve their daily activities (such as leisure, vacation, walks, work, hunting), significantly reduce their expenses (treatment costs, veterinary costs), feel significantly less emotional stress (guilt, powerlessness, sadness, regret, fear, anger, disgust, rage, frustration), and experience significantly less impact of their dogs’ illness on their relationships with family members and friends, compared to owners of dogs not treated with ASIT (questions 11, 12, 13, and 15, respectively).

A statistical comparison between the AG and the CG before treatment showed a significant difference in question 11 (Figure 4). It was found that the owners of the AG dogs were more disturbed by their dog’s disease in their usual activities (such as leisure, vacation, walks, work, hunting, etc.) than the owners of the CG dogs. Since this determinant of quality of life improved significantly after ASIT compared to the CG (Figure 3), the result is considered relevant to the treatment effect.

## 4. Discussion

The present study is the first to evaluate the quality of life of allergic dogs treated with ASIT and their owners. Significant differences were found between the quality of life of dogs and owners before and after ASIT, thus achieving the primary outcome of this study. Significant differences were found between the quality of life of dogs not treated with ASIT and that of dogs treated with ASIT, so the secondary outcome of this study was achieved.

Canine atopic dermatitis is not curable in most cases and is one of the three skin diseases that most affect the quality of life of dogs and their owners (after sarcoptic mange and pododermatitis). The disease causes dogs to be highly itchy, and when associated with skin infections, treatment becomes complicated and expensive [2,7]. The results of a recent study show that the quality of life of allergic dogs is affected mainly by behavioral changes, disturbed activities, sleep, and treatment effort. Owners’ quality of life was affected mainly by concerns about costs, loss of time, emotional stress, and physical exhaustion [2].

Therefore, the main goal of CAD management is to maintain the quality of life of allergic dogs and their owners, which should be as comparable as possible to that of healthy dogs. Every effort should be made to control pruritus and secondary skin infections. Regular measures to improve skin and coat hygiene are mandatory, as is the use of antipruritic medications when necessary [7]. Allergen-specific immunotherapy is unique among available therapies in that it interferes with the immune response of the allergic dog in that it can provide longer-term symptomatic relief compared with symptomatic treatments [15,16,17,18]. Our results show that dogs treated with ASIT have a better quality of life than dogs not treated with ASIT and may help to convince owners and clinicians of this type of treatment, which is very effective and safe in most cases [15,19,20,21,22,23].

The quality of life of the dogs and their owners who participated in the study was significantly better under ASIT than before treatment for all factors except shampooing, treatment with tablets, and ear cleaning, which presented a similar burden to the dogs and their owners as before ASIT. These results are comparable to some extent with the results of Noli’s study, which showed an improvement in the quality of all aspects of life of allergic dogs as a result of treatment, except for the burden of maintenance therapy. The study included 23 atopic dogs but differed in that it was prospective, and the duration of treatment ranged from 2 to 8 months, with an average of 3.5 months [2].

Compared with dogs not treated with ASIT, the ASIT-treated dogs in our study were significantly less disturbed at mealtimes (i.e., had better appetites and scratched less at mealtimes) and caused significantly less physical discomfort to their owners (due to unpleasant odor, the impression of a dirty apartment). Their owners were able to significantly improve their daily activities (such as leisure, vacation, walks, work, hunting), feel significantly less emotional stress (less guilt, powerlessness, sadness, regret, fear, anger, disgust, frustration), and experience less influence of their dogs’ disease on their relationships with family members and friends. They were able to significantly reduce their expenses (treatment costs, veterinary costs). These results are comparable to those of other authors. Recent research has shown that the areas of dogs’ lives most affected by chronic skin disease are behavioral and mood changes, play and work activities, and treatment exposure [2,24].

Based on the results, much of the long-term improvement in the quality of life of the dogs in this study can be attributed to ASIT treatment. Symptomatic antipruritic therapy cannot affect the course of CAD when discontinued. Symptoms recur upon contact with the triggering allergens [7]. In ASIT, a switch from the canine Th2 cell immune response to the Th1 cell immune response has been demonstrated, with an increase in specific IgG concentrations and a decrease in specific IgE concentrations compared to the pre-treatment period [25,26,27,28]. This mechanism for ASIT can relieve symptoms for prolonged periods. Thus, the need for symptomatic drugs is less significant, as is the burden of drug administration, as well as veterinary visits and costs.

Another improvement in the quality of life of dogs treated with ASIT may be attributed to the fact that ASIT has fewer long-term side effects compared with symptomatic antipruritic treatment. Adverse side effects of ASIT are expected in 2.4% of treated dogs, and, in most cases, immunotherapy can be continued after dose adjustment [29]. This was also the case in our ASIT-treated dogs. No serious side effects were observed. On the contrary, side effects can occur in up to 4.9% of cases when methylprednisolone is used for 31 days [30] and to a much lesser extent with oclacitinib or cyclosporine, but therapy with the latter two can be very costly, especially in large breed dogs [8,31]. However, the decision to perform ASIT is still that of the owner, following a discussion of its efficacy and cost, as was the case for the dogs in our study. Owners were similarly educated about treatment options, and dogs received symptomatic therapy when needed, but CG did not opt for ASIT, whereas AG did.

At the end of the discussion, some difficulties regarding the conduct of this study should be mentioned. The power of the study was calculated in advance, so the minimum number per group (i.e., 18 dogs) was known. This number was difficult to achieve at the CG, and sometimes multiple phone calls and invitations by mail were required. The response was much better in the ASIT group. This could be due to the fact that the lead physician (author of the study) had communicated more frequently with these owners in the past because of regular ASIT follow-ups. Owners were offered ongoing service in the form of prompt advice on the adverse effects of immunotherapy and dose adjustment if necessary. Regular communication with owners of allergic dogs seems to be important, as an experienced clinician can provide much support. The importance of communication has been recognized by other authors, as this improves both dog owner compliance [3] and the positive effect of ASIT [12]. In our study, it was indeed the case that the majority of owners of CG rarely sought additional advice before being invited to participate in the study.

## 5. Limitations

It should be noted that this is a retrospective study. This format may have certainly influenced the results, as the owners filled out their questionnaires based on their recollections rather immediately after treatment was finished. Some of the owners did not complete their questionnaires until years after the completion of immunotherapy. There is a possibility that some of the events were forgotten by the owners. On the other hand, long-term remissions after the cessation of immunotherapy might have been observed in this way. However, to the best of the authors’ knowledge, this is the first publication on the use of quality-of-life questionnaires in allergic dogs treated with ASIT compared to allergic dogs not treated with ASIT.

## 6. Conclusions

Based on our results, we were able to demonstrate a significant improvement in the quality of life of dogs and their owners following allergen-specific immunotherapy, thus determining the primary outcome of the study. While symptomatic therapy improved the quality of life with respect to some factors for some dogs within our control group, additional ASIT improved the quality of life in all descriptions, except for the burden of maintenance therapy. We also found significant differences in the quality of life of dogs treated with ASIT compared with the dogs not treated with ASIT. Therefore, we determined the second outcome of the study. Nevertheless, a large-scale prospective study is needed to confirm these data.

## Figures and Tables

**Figure 1 vetsci-10-00072-f001:**
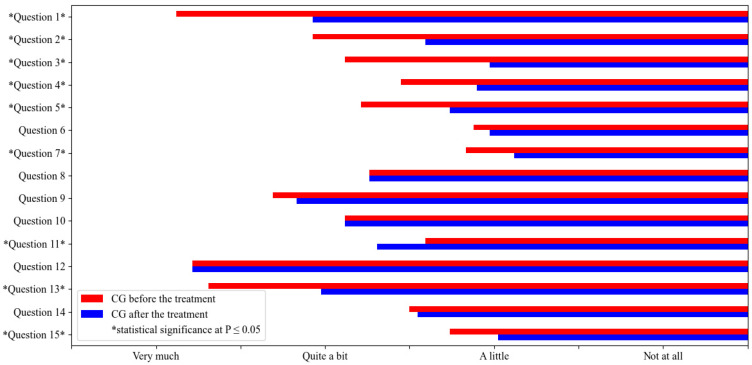
Statistical comparison of the quality of life of dogs, not treated with ASIT (CG) before the treatment (QoL1 CG) and after the treatment (QoL2 CG). The mean values are represented in the figure and statistical significance at *p* ≤ 0.05 is marked with *.

**Figure 2 vetsci-10-00072-f002:**
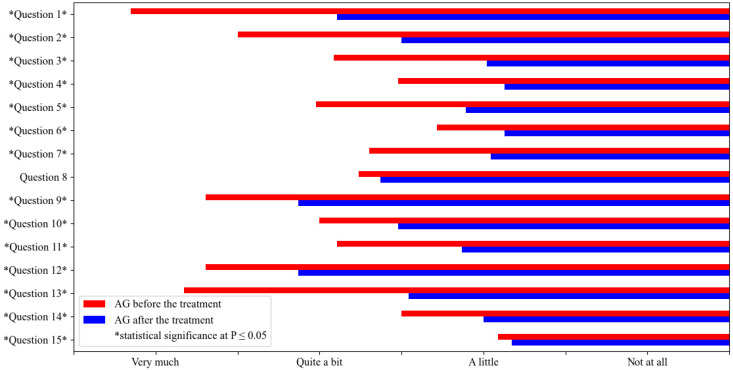
Statistical comparison of the quality of life of ASIT—treated dogs (AG) before the treatment (QoL1 AG) and after the treatment (QoL2 AG). The mean values are represented in the figure and statistical significance at *p* ≤ 0.05 is marked with *.

**Figure 3 vetsci-10-00072-f003:**
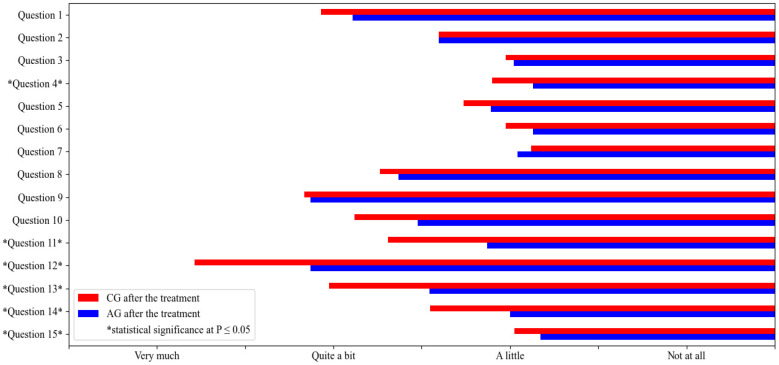
Statistical comparison of the quality of life of ASIT—treated dogs after the treatment (QoL2 AG) and dogs, not treated with ASIT, after the treatment (QoL2 CG). The mean values are represented in the figure and statistical significance at *p* ≤ 0.05 is marked with *.

**Figure 4 vetsci-10-00072-f004:**
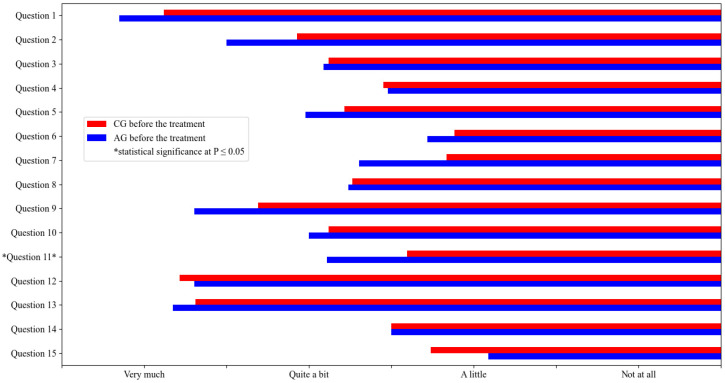
Statistical comparison of the quality of life of ASIT—treated dogs before the treatment (QoL1 AG) and dogs, not treated with ASIT, before the treatment (QoL1 CG). The mean values are represented in the figure and statistical significance at *p* ≤ 0.05 is marked with *.

**Table 1 vetsci-10-00072-t001:** Responses of the owners of dogs not treated with ASIT (control group; CG, No. = 21) before and after the treatment.

No.	Question	B Mean ± St.dev.	A Mean ± St.dev	*p* Value
1.	Severity of disease	3.381 ± 0.722	2.571 ± 0.583	Student 0.0000001
2.	Behaviour/mood influence	2.571 ± 1.003	1.905 ± 0.683	Wilcox 0.0002
3.	Sleep disruption	2.381 ± 1.045	1.524 ± 0.732	Wilcox 0.0003
4.	Meals disruption	2.05 ± 1.071	1.6 0.735	Wilcox 0.003
5.	Playing/working disruption	2.286 ± 0.933	1.762 ± 0.75	Student 0.0002
6.	Social relationship disruption	1.619 ± 0.785	1.524 ± 0.732	Wilcox 0.157
7.	Change of habits	1.667 ± 0.836	1.381 ± 0.575	Wilcox 0.014
8.	Therapies	2.238 ± 1.109	2.238 ± 0.868	Student 1.0
9.	Time loss	2.81 ± 0.906	2.667 ± 1.039	Student 0.19
10.	Physical exhaustion	2.381 ± 0.898	2.381 ± 1.045	Student 1.0
11.	Family activities disruption	1.905 ± 0.921	2.190 ± 1.096	Student 0.01
12.	Expenditure influence	3.286 ± 0.825	3.286 ± 0.765	Wilcox 1.0
13.	Emotional distress	3.19 ± 0.794	2.524 ± 1.052	Wilcox 0.0005
14.	Physical uneasiness	2.0 ± 0.926	1.952 ± 0.898	Student 0.329
15.	Family relationship influence	1.762 ± 0.971	1.476 ± 0.732	Wilcox 0.0339

B = before the treatment, A = after the treatment. The values were obtained by declaring a number for each answer (Not at all = 1, A little = 2, Quite a bit = 3, Very much = 4) and then the mean number and standard deviation = St.dev. for each question were calculated.

**Table 2 vetsci-10-00072-t002:** Responses of the owners of dogs treated with allergen-specific immunotherapy (AG, No. = 46) before and after the treatment.

No.	Question	B Mean ± St.dev.	A ± Mean ± St.dev	*p* Value
1.	Severity of disease	3.652 ± 0.633	2.391 ± 0.966	Student < 0.0000001
2.	Behaviour/mood influence	3.000 ± 0.834	2.000 ± 0.834	Wilcox < 0.0000001
3.	Sleep disruption	2.413 ± 0.946	1.478 ± 0.827	Wilcox < 0.0000001
4.	Meals disruption	2.022 ± 1.053	1.370 ± 0.703	Wilcox 0.000006
5.	Playing/working disruption	2.522 ± 0.926	1.609 ± 0.82	Student < 0.0000001
6.	Social relationship disruption	1.783 ± 0.930	1.370 ± 0.733	Wilcox 0.000013
7.	Change of habits	2.196 ± 1.055	1.457 ± 0.877	Wilcox 0.0000008
8.	Therapies	2.261 ± 0.895	2.130 ± 1.013	Student 0.06
9.	Time loss	3.196 ± 0.797	2.630 ± 0.941	Student < 0.0000001
10.	Physical exhaustion	2.500 ± 0.853	2.022 ± 0.944	Student < 0.0000001
11.	Family activities disruption	2.391 ± 0.872	1.630 ± 0.817	Wilcox < 0.0000001
12.	Expenditure influence	3.196 ± 0.924	2.630 ± 0.941	Student < 0.0000001
13.	Emotional distress	3.326 ± 0.886	1.957 ± 0.908	Student < 0.0000001
14.	Physical uneasiness	2.000 ± 1.083	1.500 ± 0.828	Wilcox 0.00001
15.	Family relationship influence	1.413 ± 0.768	1.326 ± 0.753	Wilcox 0.0455

B = before the treatment, A = after the treatment. The values were obtained by declaring a number for each answer (Not at all = 1, A little = 2, Quite a bit = 3, Very much = 4) and then the mean number and standard deviation = St.dev. for each question were calculated.

## Data Availability

Not applicable.

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
