# Peer review of "Quality of Life of Allergic Dogs Treated with Allergen-Specific Immunotherapy—A Retrospective Study"

_vetsci, 2023, doi:10.3390/vetsci10020072_

Round 1

Reviewer 1 Report

This is a review of the manuscript entitled Quality of life of allergic dogs, treated with antigen specific immunotherapy – a retrospective study.

The main goal of this retrospective study was to evaluate the impact of an allergen-specific immunotherapy (ASIT) on the quality of life of dogs suffering from atopic dermatitis as well as on their owners.

This is an interesting field where several studies have been published in veterinary dermatology. This is a clinically relevant study that adds new data to encourage owners of atopic dogs to invest time and money in ASIT. While the design itself seems adequate (considering that a retrospective has its own limitations), there is many information missing, especially regarding treatments and family/environmental context, to better interpret the results. That said, there are so many factors that can influence the course of atopic disease that it may be difficult to draw clear conclusions about the long-term effectiveness of any therapy.

Here are some comments to authors.

Title: Allergen-specific immunotherapy is more appropriate than antigen-specific…

Both expression are used in the text. The immunotherapy does not contain specific antigens, but allergen extracts. I suggest to standardize the text and change «antigen» for «allergen».

Introduction:

Line 10: Since most animals are polysensitized and ASIT usually contains more than one allergen, I suggest to use the plural form: … causative allergens.

Lines 43-45 : I don’t exactly understand the meaning of this affirmation. I would appreciate if the author can clarify how the questionnaire can improves owner’s compliance. In fact, a questionnaire does not constitute recommendations or a therapy protocol.

Line 65: Since most animals are polysensitized and ASIT usually contain more than one allergen, I suggest to use the plural form: … quantities of allergen extracts to an …

Line 66: …. causative allergens (plural form)

Line 68: The author suggests that the ASIT can modify or reverse the pathogenesis of the disease. Actually, it’s not exactly the pathogenesis of the disease that is modified by the ASIT.  The ASIT will influence the course of the disease by improving tolerance to triggering allergens.

M&M:

Lines 78-81: The author precise that various symptomatic drugs were allowed for both groups. In order to better interpret the results, it would be important to precise which treatment were allowed (antipruritic? antimicrobial? antiparasitic?).

Lines 103-109: Considering that ASIT will require several months to be effective, it would be important to add information regarding the duration of maintenance therapy required to be enrolled. In addition, there are various published therapeutic protocols for ASIT. I suggest that the author add information regarding which protocol was used, and whether the same desensitization protocol was used for all enrolled dogs.

Results:

Lines 139-141: Please clarify the first sentence. The ASIT has been stopped by the owners or finished according to the protocol? The dogs for which the ASIT were discontinued, are they included in the AG or CG? Please clarify.

General comments:  In the results section, there is no mention of the treatments received by the dogs enrolled in the two groups. This is information that can significantly affect the results of this study. For example, if most of the dogs in the CG group are receiving antihistamines while those in the AG group are being treated primarily with glucocorticoids, it is obvious that the AG group will have better results, ASIT or not. Without this information, it is really difficult to properly interpret the data. We can also have the same comments about other factors that influence pruritus. The author mentions that ectoparasites were ruled out before the diagnosis of atopic dermatitis. However, there is a gap between the time of diagnosis and the study. It would be important to clarify whether all these dogs are regularly treated with an antiparasitic. Ectoparasites are certainly one of the possible causes of ASIT "failure".

Lines 147-150: Please precise how many owners responded to both questionnaires. In fact, reading the table, it seems that not all owners answered both questionnaires.

Tables 1 and 2: There are many points to be improved :

              -not easy to follow the data; I suggest to present the table differently

              -many numbers are not aligned properly

              -for many questions, the total number of answers are less than or even greater than the number of dogs. If less number means that not all owners answered the question, it is very important to make that clear. But how can it be more than the number of dogs? E.g. Table 1 (n=21 dogs) Qu 2: 25 answers for A (after ASIT)

Figures 1, 2, 3, 4: I suggest to change de color of the bars. It is difficult to distinguish the green from the blue bars. These colors are too confluent.

Discussion:

Lines 210-216: The purpose of this study is not the pathogenesis of the disease: this paragraph does not add a value to the discussion. I suggest to summarize the idea. Line 215 is a result, add nothing to the discussion.

Lines 226-235: Although this review on the hypothesis to explain how ASIT works is interesting, it is not relevant for the current study: it’s not a review article. In the discussion, the results of the study should be discussed. I suggest to delete this section or summarize it.

Line 242-243: Which results are understandable?

Line 244: «…. improved rapidly and immunotherapy was completed in less than 10 months.» This part of this sentence is confusing. What does the author means by «completed»? The immunotherapy is stopped if we consider the therapy ineffective, the animal had side effects, or if the owner has decided to stop it. But if there are benefits, the immunotherapy is continued. It’s not a therapy with an end point in time, so we can not say that the treatment is «completed».

Lines 226-271: This section of the discussion is all about the efficacy of ASIT. Again, the purpose of this study is not to assess the efficacy of ASIT, but QoL when this therapy is used. If the author would like to discuss these interesting data, I suggest to present these data differently, perhaps in table, or to modify the title and add a second objective being the efficacy of ASIT. Otherwise, all this section is irrelevant in that context.

Overall, the discussion does not really focus on the results. The author should expand on this section and discuss the treatments received and how this may have influenced the results. The author can also comment on the content of the ASIT. For example, was the quality of life better in dogs receiving ASIT containing only mites?  It would also be interesting to know more about the dog's environment. For example, we might think that dogs living with other dogs might have improved more rapidly the QoL because of the social interactions. Is there a difference in perception between owners if it is a woman or a man who answered the questionnaire? Was it the same person in the family who answered the first and second questionnaires? Is there a difference in the perception of the dog's quality of life if the dog lives with a single owner or in a family? There are many factors that may influence the interpretation of the quality of life of these dogs that are not addressed in this study. In the context of this study, it would have been even more interesting to elaborate on these aspects in the discussion, rather than the theory on how ASIT influences the course of the disease.

Author Response

Reply to the reviewer 1:

The author thanks the two reviewers for their correct and useful comments, which undoubtedly improved the manuscript. The comments have been respected as much as possible, and the corrections are explained below.

Reviewer 1:

Title:

Reviewer: “Allergen-specific immunotherapy is more appropriate than antigen-specific…”

The whole text has been standardized and changes of «antigen» for «allergen» have been done.

Simple Summary and Introduction:

Line 10: the plural form has been used: … “causative allergens”

Lines 47-50: Reviewer: »Lines 43-45 : I don’t exactly understand the meaning of this affirmation. I would appreciate if the author can clarify how the questionnaire can improves owner’s compliance. In fact, a questionnaire does not constitute recommendations or a therapy protocol”

An explanation has been added, hereby copy-pasted. “By being able to show improvement in quality of life during treatment (by comparing owners' responses at different times during therapy), the clinician can more easily gain owners' devotion«.

Line 68: the plural form has been used as suggested: … “quantities of allergen extracts to an”

Line 69: the plural form has been used”…. “causative allergens”

Line 70-72: Reviewer: “The author suggests that the ASIT can modify or reverse the pathogenesis of the disease. Actually, it’s not exactly the pathogenesis of the disease that is modified by the ASIT.  The ASIT will influence the course of the disease by improving tolerance to triggering allergens.” 

Response: "Pathogenesis" was replaced by "immune response": “It is the only current treatment for allergy that can alter or reverse at least part of the immune response in this condition, thereby both relieving clinical symptoms and preventing disease progression.«….

M&M:

Lines 83-85: Reviewer: “The author precise that various symptomatic drugs were allowed for both groups. In order to better interpret the results, it would be important to precise which treatment were allowed (antipruritic? antimicrobial? antiparasitic?).”

The sentence has been added:The drugs that were allowed to be used in both groups were topical and systemic antipruritics, antimicrobials, and antiparasitic treatments.

Lines 110-116: “Reviewer: Considering that ASIT will require several months to be effective, it would be important to add information regarding the duration of maintenance therapy required to be enrolled. In addition, there are various published therapeutic protocols for ASIT. I suggest that the author add information regarding which protocol was used, and whether the same desensitization protocol was used for all enrolled dogs”.

The explanation has been added: All participating dogs were prescribed the same desensitization protocol (Nextmune official instructions). Briefly, the protocol consisted of s/c injections of allergen mixtures increasing from 0.2 ml to 1.0 ml, with injections started every 2 weeks for 4 times, continued every 3 weeks for 2 times, and maintained every 4 weeks. Individual adjustments to the dose were possible if the physician determined intolerance. The duration of therapy under study was not limited as long as reactions were cleared.

Results:

Lines 152-155: Reviewer: “Please clarify the first sentence. The ASIT has been stopped by the owners or finished according to the protocol? The dogs for which the ASIT were discontinued, are they included in the AG or CG? Please clarify.”

Response: The explanation was added: »Discontinuation of treatment was by veterinary or owner decision because the clinical condition had improved in 5 of 12 dogs (41.7%) and worsened in the others (15.2%). The dogs in which ASIT was discontinued regardless of response were included in the AG.

Reviewer: »General comments:  In the results section, there is no mention of the treatments received by the dogs enrolled in the two groups. This is information that can significantly affect the results of this study. For example, if most of the dogs in the CG group are receiving antihistamines while those in the AG group are being treated primarily with glucocorticoids, it is obvious that the AG group will have better results, ASIT or not. Without this information, it is really difficult to properly interpret the data. We can also have the same comments about other factors that influence pruritus. The author mentions that ectoparasites were ruled out before the diagnosis of atopic dermatitis. However, there is a gap between the time of diagnosis and the study. It would be important to clarify whether all these dogs are regularly treated with an antiparasitic. Ectoparasites are certainly one of the possible causes of ASIT "failure".

Response to general comment: the author essentially agrees with the above general comments. Of course, each of the concurrent treatments (as well as many other variables) can affect the response to therapy. These are variables that the researcher could not control in this retrospective study, and therefore it would be pointless to include or discuss them. Variables such as: concurrent treatments, environmental variables, gender, age, sensitivity to different allergens, other animals present, etc. could only be controlled in prospective studies. It is important that the owners of the dogs were equally educated about the disease and treatment options, and symptomatic medications were used as needed and according to treatment guidelines. At the time of the study, only a limited number of effective antipruritics were available (essentially prednisolone /methylprednisolone and oclacitinib), and this will be addressed in the revised manuscript. The owners of CG and AG had the support of the same physician (the author). Finally, speaking of years of treatments, it is really impossible to recognise every exacerbation. Concomitant treatments may be prescribed by different physicians and are not documented in a single file in one place (veterinary practice). The author would like to point out that the efficacy of ASIT per se was not the outcome of the study, as it is already well established in the literature. The outcome of the study was to compare quality of life on a long-term basis.
The author has provided additional explanation at many points in the manuscript to support what has been written above.

Lines 163-167: Reviewer: “Please precise how many owners responded to both questionnaires. In fact, reading the table, it seems that not all owners answered both questionnaires”

Response: Explanation was added: “Owners were instructed to answer QoL1 first and then QoL2. For each question on QoL1, notice "before treatment" was written, and for each question on QoL2, notice "after treatment" was written. Only one owner answered the questionnaires for each dog, and only the dogs of owners who completed all questions and returned both questionnaires were included in CG and AG.«

Discussion:

Lines 230-235: Reviewer: “The purpose of this study is not the pathogenesis of the disease: this paragraph does not add a value to the discussion. I suggest to summarize the idea. Line 215 is a result, add nothing to the discussion.”

Response: Original paragraph..... “Canine atopic dermatitis (CAD) is a disease that is not curable in the majority of cases [8]. The diagnosis of CAD is based on meeting clinical criteria and ruling out other possible causes with similar clinical signs [11]. Since IgE levels may be elevated in CAD-affected and non-affected dogs [12] allergy testing is not used to confirm the diagnosis. Once a clinical diagnosis of CAD is made, allergy testing is performed to identify potentially causative allergens for allergen-specific immunotherapy [11]. Accordingly, in our study the test was performed on dogs from AG but not on dogs from CG.”

.......Was changed to this paragraph:

"The present study is the first that evaluated quality of life of allergic dogs, being treated by ASIT and of their owners. Significant differences were found between the quality of life of dogs and owners before and after ASIT, thus achieving the primary outcome of this study. Significant differences were found between the quality of life of dogs not treated with ASIT and dogs treated with ASIT, so the secondary outcome of this study was achieved."

Lines 236-243: Although this review on the hypothesis to explain how ASIT works is interesting, it is not relevant for the current study: it’s not a review article. In the discussion, the results of the study should be discussed. I suggest to delete this section or summarize it.

Response: Original paragraph .....“Allergen-specific immunotherapy is recommended in all allergic dogs that suffer from symptoms for more than three months per year and in whom dietary control and treatment with symptomatic medications are inadequate due to side effects of the medications used and/or lack of owner compliance with therapy [6]. It is common clinical practice to suggest ASIT in any case of CAD when the above criteria are met, and so it was with the dogs in our study. However, the decision to perform ASIT is still an individual decision of the owner. Decades of clinical experience and research data are compelling regarding the benefits of ASIT [9], but explaining the efficacy and cost of this therapy to owners is always mandatory.”

........Was changed to this paragraph:

"Canine atopic dermatitis is not curable in most cases and is one of the three skin diseases that most affect the quality of life of dogs and their owners (after sarcoptic mange and pododermatitis). The disease is highly itchy, and when associated with skin infections, treatment becomes complicated and expensive [2,7]. The results of a recent study show that the quality of life of allergic dogs is affected mainly by behavioural changes, disturbed activities, sleep, and treatment effort. Owners' quality of life was affected mainly by concerns about costs, loss of time, emotional stress, and physical exhaustion [2]."

Line 242-243: Reviewer: “Which results are understandable?”

Response: This sentence was deleted.

Line 247: Reviewer:«…. improved rapidly and immunotherapy was completed in less than 10 months.» This part of this sentence is confusing. What does the author means by «completed»? The immunotherapy is stopped if we consider the therapy ineffective, the animal had side effects, or if the owner has decided to stop it. But if there are benefits, the immunotherapy is continued. It’s not a therapy with an end point in time, so we can not say that the treatment is «completed».

Response: Agreed. The sentence was changed to: “At the time of inclusion immunotherapy have already been finished in a period ≤10 months in 12 cases (26.1%). Discontinuation of treatment was by veterinary or owner decision because the clinical condition had improved in 5 of 12 dogs (41.7%) and worsened in the others (15.2%).« This paragraph was moved to Results section.

Lines 244-294: Reviewer: “This section of the discussion is all about the efficacy of ASIT. Again, the purpose of this study is not to assess the efficacy of ASIT, but QoL when this therapy is used. If the author would like to discuss these interesting data, I suggest to present these data differently, perhaps in table, or to modify the title and add a second objective being the efficacy of ASIT. Otherwise, all this section is irrelevant in that context.”

Response: the author is grateful for this comment and has thoroughly reconstructed the discussion, focusing on the quality of life. Additional parts of the discussion have been added. Most references were retained for comparison purposes, and some new ones were added.

Reviewer:”Overall, the discussion does not really focus on the results. The author should expand on this section and discuss the treatments received and how this may have influenced the results. It would also be interesting to know more about the dog's environment. For example, we might think that dogs living with other dogs might have improved more rapidly the QoL because of the social interactions”.

Response: These concerns were addressed already above in the section of Results (Response to general comment).

Reviewer: “The author can also comment on the content of the ASIT. For example, was the quality of life better in dogs receiving ASIT containing only mites?”

Response: The author does not consider the search for a correlation between ASIT composition and quality of life to be clinically relevant. The author expects a correlation between ASIT composition and response to treatment, but this was not the purpose of this study.

Reviewer:  “Is there a difference in perception between owners if it is a woman or a man who answered the questionnaire? Was it the same person in the family who answered the first and second questionnaires? Is there a difference in the perception of the dog's quality of life if the dog lives with a single owner or in a family?”

Response: The reviewer opens here very interesting questions that cannot be answered on the basis of the results of this study. However, some of them have already been answered in previous studies. Noli's study evaluated the same QoL questionnaire that was used in our study, and found that the sex, age, or education level of the owner did not affect QoL outcomes (Noli et al., Part 2). And yes, in our study, QoL1 and QoL2 were answered by the same person. Thank you to the reviewer's comment, this sentence was included in the M&M section.

Reviewer:”There are many factors that may influence the interpretation of the quality of life of these dogs that are not addressed in this study. In the context of this study, it would have been even more interesting to elaborate on these aspects in the discussion, rather than the theory on how ASIT influences the course of the disease.”

Response: The author agrees that allergic diseases in humans and animals represent a large, very interesting and endless field of research. The author has been involved with CAD and FASS, clinically, in education, and to some extent as a researcher, for more than 30 years. Personally, the author has had the opportunity to witness a great development of knowledge in this field, but on the other hand, she has also had the opportunity to witness many misdiagnoses and irrelevant treatments by general practitioners on a daily basis. The training of these physicians is very important. Unfortunately, many of them still do not know the unique effect and high positive response of ASIT. Therefore, the author believes it is important to promote this therapy, which is directly related to the quality of life of all those unfortunate dogs and cats who suffer all their lives from chronic otitis externa, recurrent pyodermas, constant itching and the side effects of glucocorticoids, only because many physicians are still unaware of the effectiveness of ASIT. The author believes that the obvious improvement in quality of life with ASIT may help convince physicians to use this therapy more frequently.

Reviewer 2 Report

The study of Kotnik was on the quality of life of dogs diagnosed with canine atopic dermatitis. Although the topic is exciting and relevant, I have some major comments which could improve the manuscript/study.

MAJOR:

1) Table 1, 2. Both tables expand over 4 pages, making data very unclear! Since the questionnaire is standardized, I would move both tables into a Supplement and keep only a summarized version. For example, the first question could be shortened to “disease disturbance”, the second to “impact on general behavior,” etc. For each question, instead of a separate number of answers, I would keep a single number (e.g., means/medians + deviation/percentiles) for before/after treatments.

2) Table 1,2. Not all abbreviations are explained: ASIT, SW, Norm, NonNor, Wilcox, etc. The last two columns on statistics are not clear without quite some thinking.

3) Figure 1-4. What are the units of bars? Means, medians,...? Also, not all abbreviations are explained in the figure caption (ASIT, CG, QoL,...).

4) Figures 1-4. Currently, absolute comparisons between groups are made: QoL CG before vs. after treatment (Figure 1), QoL AG before vs. after T (Figure 2), QoL after T CG vs. AG (Figure 3), and QoL before T CG vs. AG (Figure 4). Since there were some significant differences between AG and CG before treatment (lines 192-193), this approach is weak. Please make a differential comparison of QoL CG (before - after T) vs. QoL AG (before - after T). In other words, the suggested approach compares the differences in QoL before and after the treatment between both groups (CG, AG). This will offer much better insight into ASIT impact and maybe also answer your hypothesis (20%-better QoL of AG?). 

5) Discussion is very long, mostly writing about CAD-related topics in general. a) Please move the general text into the introduction (e.g., lines 208-242). b) The actual discussion related to your and other results should be expanded (currently, it only covers lines 284-294).

6) M&M, lines 78-81. Do you know why some dogs received ASIT and others did not? For example, was that predominantly the owners’ decision due to expected costs? This could be an important factor when interpreting differences between both dog groups.

7) M&M, line 129. Why was the significance level raised to 0.1 and not left at 0.05 as with initial tests?

MINOR:

1) Abstract, line 12. I suggest adding information that also dogs in the control group were allergic. Currently, this information is not evident from the abstract text.

2) Introduction, lines 55-60. I suggest adding that lokivetmab can be very convenient due to only monthly administration. Additionally, antihistamines do not to be stressed since they are not very effective in managing CAD.

Author Response

The author thanks the  reviewer  for his correct and useful comments, which undoubtedly improved the manuscript. The comments have been respected as much as possible, and the corrections are explained below.

  • Table 1, 2. Both tables expand over 4 pages, making data very unclear! Since the questionnaire is standardized, I would move both tables into a Supplement and keep only a summarized version. For example, the first question could be shortened to “disease disturbance”, the second to “impact on general behavior,” etc. For each question, instead of a separate number of answers, I would keep a single number (e.g., means/medians + deviation/percentiles) for before/after treatments.

 Response: Done. The original tables were moved to a supplement, and summary versions were created with individual numbers (e.g., mean +/- deviation) for the before / after treatments. The abbreviations of the questions were taken from the Noli's article (in which the authors actually evaluated and validated these questionnaires) to facilitate comparison of our results with this study.

2) Table 1,2. Not all abbreviations are explained: ASIT, SW, Norm, NonNor, Wilcox, etc. The last two columns on statistics are not clear without quite some thinking.

 Response: Done. The original tables were moved to a supplement, and summary versions were created with all abbreviations explained.

3) Figure 1-4. What are the units of bars? Means, medians,...? Also, not all abbreviations are explained in the figure caption (ASIT, CG, QoL,...).

Response: Done. Units are Means and explained under the figures.

4) Figures 1-4. Currently, absolute comparisons between groups are made: QoL CG before vs. after treatment (Figure 1), QoL AG before vs. after T (Figure 2), QoL after T CG vs. AG (Figure 3), and QoL before T CG vs. AG (Figure 4). Since there were some significant differences between AG and CG before treatment (lines 192-193), this approach is weak. Please make a differential comparison of QoL CG (before - after T) vs. QoL AG (before - after T). In other words, the suggested approach compares the differences in QoL before and after the treatment between both groups (CG, AG). This will offer much better insight into ASIT impact and maybe also answer your hypothesis (20%-better QoL of AG?). »

Answer: we actually performed the proposed test (see Figure 5 below) and can see that AG and CG differ before treatment only in question 11. On this question, AG performed worse than CG before treatment, but on the same question after treatment, AG performed better than CG. A test at the 0.05 significance level was also performed (as suggested in comment #7 below), which revealed the only difference in question #11.
Therefore, the proposed statistical test (shown here as Figure 5) was not included in the article but everything is explained in the results, supported by Figure 4 (please see attached file).

5) Discussion is very long, mostly writing about CAD-related topics in general. a) Please move the general text into the introduction (e.g., lines 208-242). b) The actual discussion related to your and other results should be expanded (currently, it only covers lines 284-294).

Answer: the author thanks both reviewers for his, roughly the same, comment. It was done, the discussion was thoroughly reconstructed.

6) M&M, lines 78-81. Do you know why some dogs received ASIT and others did not? For example, was that predominantly the owners’ decision due to expected costs? This could be an important factor when interpreting differences between both dog groups.

Answer: The author finds this question interesting. Since the author has over 25 years of experience with ASIT, she knows several reasons why it is not allways done: It is owners' decisions because of the price, sometimes they are reluctant to give injections, sometimes they are of the opposite opinion of some practitioners... some owners are simply looking for a quick relief of symptoms that we cannot promise with ASIT, etc. Since this was a retrospective study, the exact reasons for owners' participation in our CG were not known (this information was not collected in a questionnaire). The focus was on quality of life. Nevertheless, a brief discussion was held between Lines 290-294: 

“However, the decision to perform ASIT is still an individual decision of the owner after discussing its efficacy and cost, and this was the case for the dogs in our study. Owners were similarly educated about treatment options, and dogs received symptomatic therapy when needed, but CG didn't opt for ASIT, whereas AG did”.

7) M&M, line 129. Why was the significance level raised to 0.1 and not left at 0.05 as with initial tests?

Response: Corrected in a revised manuscript. Significance level in the revised manuscript is set at 0.05 for all statistics and so written in M&M, lines 133-140.

1) Abstract, line 28. I suggest adding information that also dogs in the control group were allergic. Currently, this information is not evident from the abstract text.

Answer: Done in Line 28: »CG consisted of 21 dogs with CAD and AG of 46 dogs with CAD«.

2) Introduction, lines 59-62. I suggest adding that lokivetmab can be very convenient due to only monthly administration. Additionally, antihistamines do not to be stressed since they are not very effective in managing CAD.

 Response: Agreed, in the review version antihistamines are not stressed any more. About Lokivetmab : the drug was not available at our market at the time of the survey, therefore further discussion is not relevant.

Round 2

Reviewer 1 Report

The explanations and corrections made are satisfactory.

Thank you for taking the comments into account.

Reviewer 2 Report

I do not have any further comments.